# Motives of Students for or against the Practice of Physical Exercise and How They Are Related with the Climate in Physical Education Classes

**DOI:** 10.3390/ijerph18168348

**Published:** 2021-08-06

**Authors:** Sebastián Peña-Troncoso, Laura Espinoza-Sánchez, Claudio Hernández-Mosqueira, Sergio Toro-Arévalo, Jaime Carcamo-Oyarzun, Gustavo Pavez-Adasme, Hugo Velásquez-González

**Affiliations:** 1Instituto de Ciencias de la Educación, Universidad Austral de Chile, Valdivia 5110566, Chile; sebastian.pena@uach.cl; 2Facultad de Educación y Cultura, Universidad SEK, Santiago 7520317, Chile; lauraisabelespinozasanchez@gmail.com; 3Departamento de Educación Física, Deportes y Recreación, Universidad de La Frontera, Temuco 4780000, Chile; jaimecarcamo@outlook.com; 4Facultad de Ciencias Médicas, Universidad de Santiago, Santiago 7750000, Chile; seatoro@gmail.com; 5Grupo de investigación AFSYE, Universidad Adventista de Chile, Chillan 3780000, Chile; gustavopavez@unach.cl; 6Carrera Ciencias del Deporte, Universidad Santo Tomas, Valdivia 5090000, Chile; hugovelasquez@santotomas.cl

**Keywords:** motivation, exercise, physical education, students, adolescent

## Abstract

The main aim of the study was to investigate the reasons that motivate secondary school pupils to practise physical exercise, and how these motives are related to their perception of the climate in physical education classes. Participants: The sample consisted of 448 subjects, 36.8% girls and 63.2% boys, aged between 16 and 19 years (M = 17.61; SD = 0.96). The methodology was a descriptive cross-sectional study. Two instruments were applied: self-reporting by the pupils of their motives for practising physical exercise, and a questionnaire on their attitudes towards teachers’ behaviour and the physical education programme. The results showed that the strongest motive for the practice of physical exercise was “Prevention and positive health” (M = 5.29; SD = 1.45). In conclusion, there is a weak correlation among the pupils’ motives for practising physical exercise and their perception of the climate in class.

## 1. Introduction

Physical inactivity or lack of exercise in children and adolescents has increased exponentially in recent years around the world, becoming a serious public health problem [1]. One of the explanations may lie in the lack of motivation to practise physical exercise [2]. From this perspective physical education (PE) class plays a fundamental role in schools [3], increasing pupils effort to learn and achieve success in school [4], and strengthening their motivation [5], which in turn is linked to preventing them from dropping out of sports activities [6,7]. The work of the PE teacher is of fundamental importance in schools [8], since it promotes autonomy and the motivation to practise physical exercise among pupils, especially at an age where social influences and socio-demographic aspects affect motivation [9,10,11], the more so as parents attach less importance to PE classes than to other subjects like mathematics, language and English [12]. It is therefore fundamental to generate motivations in pupils which recognise the importance of physical exercise as an element of living well [5,13], thus reducing the probability that they will abandon the practice of exercise and an active life when they leave school [14]. One of the fundamental stages for the consolidation of motivation to practice physical exercise is adolescence, since this is when pupils start to lose motivation and interest in educational contents related to physical exercise [15].

Another important aspect of the motivation to practice sport is the association with pupils academic results in physical education [16,17,18], since the learning achieved in class is generally attributed to high motivation in the pupil and an environment that favours this motivation. Pupils who state that they do not like PE classes perceive a stronger motivational climate oriented towards performance, while those who do enjoy the classes develop a better climate for learning, and even greater orientation towards work and the self [19]. From this perspective, it is essential to adapt teaching practices to the individual needs of each pupil, seeking to create a permanent classroom climate which will allow them to enjoy a motivating, affective learning experience [20]. Various authors [3,21] have shown that orientation to a task and perception of a positive motivational climate for learning are positive predictors of practising sport outside school. Likewise, affective processes and motivation provide a formative influence in self-cognition and perception, and in learning in general [22,23]. It is therefore a great challenge to help to identify pupils perceptions of the climate in PE classes, and reasons for the lack of motivation in pupils to practise exercise both in class and out of school, since it must not be forgotten that exercise out of school is also an important complement to learning [24].

Various studies [25,26,27] point to different benefits provided by physical activity in pupils physiological, psychological, and social dimensions. However, we must go beyond the technocratic paradigm attributed to physical education in schools and imagine a class that promotes living well from the pupils multi-dimensional nature, giving them critical, creative, existential possibilities and positioning skills for their own lives, towards a sustainable, healthy life [8].

Bearing this in mind, it is necessary to promote a line of research in which little evidence has been published in Chile. Therefore, the main aim of this study was to investigate the reasons that motivate secondary school pupils to practise physical exercise, and how these motives are related with their perception of the climate in physical education classes.

## 2. Materials and Methods

The design of the study was quantitative, non-experimental, descriptive and cross-sectional, with a single measurement [28], through the application of two validated, reliable questionnaires.

### 2.1. Participants

A total of 448 pupils took part in the study (36.8% girls and 63.2% boys) from eighth grade primary school and first and second grade secondary school (equivalent to grades 11 and 12 in high school) in schools in the District of La Florida, Metropolitan Region, Chile. They were aged between 16 and 19 years (M = 17.61; SD = 0.96). Convenience sampling was used [29] to obtain representation from municipal and direct grant schools for the state of Chile. All the data were obtained after cleaning and detection of outliers (Table 1). The anonymity of all the pupils was safeguarded. Participants aged under 18 signed an informed assent and those over 18 signed an informed consent. In the case of minors, their parents or guardians signed an informed consent, allowing the pupil to participate in the investigation. In this way the study complied with the ethical criteria of the Helsinki Declaration on studies on human beings [30]. The present study was approved by the ethics committee of the Universidad Adventista de Chile (Ethics report N°2021-11). The main inclusion criteria were being enrolled in a municipal and direct grant schools, belonging to the commune of Florida, and being in the age range of 16 to 20 years. The main exclusion criterion was belonging to private schools.

### 2.2. Instruments

Motivation for physical exercise: the study used the Spanish version of the Self-report of Reasons for the Practice of Physical Exercise, AMPEF [31], created from the Exercise Motivations Inventory-2 (EMI-2) of Markland and Ingledew [32] which is of proven validity and reliability [33]. The questionnaire consists of 48 reasons for practice (items) grouped into eleven factors: (1) Body mass and body image, (2) Fun and wellness, (3) Prevention and positive health, (4) Competition, (5) Affiliation, (6) Muscular strength and endurance, (7) Social recognition, (8) Stress control, (9) Agility and flexibility, (10) Challenge and, (11) Health emergencies. The response format in the original questionnaire is a 10-point Likert-type scale, however after validation by judgement of experts it was modified to 8 points (0–7), mainly because the reviewers pointed out that since it is an even-numbered scale, on many occasions the participants are inclined to answer in the medium term (in this case 5). Finally, the maximum score of the AMPEF was 336 points.

Perception of class climate: we applied the Spanish version of the Student Attitude Toward Teacher Behaviour and Programme Content in School Physical Education questionnaire (CAPPEF) created by Gutiérrez-San Martín and Pilsa [34]. The questionnaire consists of 29 items, divided into two factors: the first measures pupils attitudes to teacher behaviour (15 items), and the second measures pupils’ attitude to the PE programme content (14 items). The maximum possible score in CAPPEF is 145 points. The response format is a 5-point Likert-type scale from (1) totally disagree, to (5) totally agree.

To check the relevance and validity of the content of the instruments, each questionnaire was subject to review by the judgement of experts [35]. Five teachers were selected as experts, each with a long academic career in physical education in Chile. Once the teachers judgement had been heard, it was decided to keep the 48 items of the AMPEF, but the Likert-type scale was reduced to 8 points, where 0 indicated total disagreement with the statement and 7 total agreement. In the CAPPEF, all 29 items were retained, with some semantic adjustment in the phrasing. Cronbach’s alpha coefficient was applied to determine the reliability of the instruments, where AMPEF obtained α 0.97 and CAPPEF α 0.93, showing that both instruments possess very good internal consistency.

### 2.3. Procedures

The anonymity of all the pupils was safeguarded. Participants aged under 18 signed an informed assent and those over 18 signed an informed consent. In the case of minors, their parents or guardians signed an informed consent, allowing the pupil to participate in the investigation. In this way the study complied with the ethical criteria of the Helsinki Declaration on studies in human beings [30]. The present study approved by the ethics committee of the Universidad Adventista de Chile (Ethics report N°2021-11-28-04-2021).

Before the surveys were applied, meetings were held with the senior management of the schools (director, general inspector, head of technical-pedagogic unit), in order to obtain permission and explain the format and objectives of the investigation. Meetings were also held with the grade teachers of the grades in which the surveys were to be applied, to settle details of the day, time and other administrative aspects of application. The surveys were applied in the period assigned to the grade councillor. The investigators were present, they explained to the pupils how to answer the survey and answered any questions.

### 2.4. Data Analysis

Descriptive analysis was carried out of the central tendency (mean) and dispersion (standard deviation) to characterise the variables. The Kolmogorov–Smirnov test was used to identify the normality of the data distribution. Non-parametric tests were applied. The Mann–Whitney U test and the Kruskal–Wallis test were used for inferential analyses. Spearman’s coefficient was used to determine correlations among the variables “Motivation for physical exercise” (AMPEF) and “Perception of climate in class” (CAPPEF). The level of significance used in all statistical analyses was *p* ≤ 0.05. SPSS version 23 software was used.

## 3. Results

This section presents the most significant results, based on the central tendency (mean) and dispersion (standard deviation) of both the sample as a whole and for the variables sex, grade, motivation for physical exercise and perception of class climate. It also presents the results of the analyses of the relations between the different variables studied.

Table 2 shows the results for the mean score (M) and standard deviation (SD) obtained for the sample, as a function of the AMPEF variables, by sex.

Table 2 shows the results for the mean score and standard deviation obtained for the sample set, as a function of sex, in the eleven factors of the AMPEF questionnaire. Considering the overall results, as well as the logical difference as a function of sex, statistically significant differences were found, with specific results in the Mann–Whitney U test z = −3.619; *p* = 0.00.

As a function of the results obtained by sex, we observe that boys place a higher value on the factors “Prevention and positive health” (M = 5.41 ± 1.46) and “Muscular strength and endurance” (M = 5.29 ± 1.49), while girls place a higher value on “Prevention and positive health” (M = 5.09 ± 1.40) and “Body mass and body image” (M = 4.65 ± 1.49).

When the factors were compared by sex, statistically significant differences were found in seven factors; competition, affiliation, muscular strength and endurance, social recognition, challenge, fun, and prevention, all with (*p* < 0.05).

Table 3 shows the results for the mean score and standard deviation obtained for the sample set, as a function of sex, in the two factors of the CAPPEF questionnaire. Considering the overall results, as well as the logical difference as a function of sex, no statistically significant differences were found, with specific results in the Mann–Whitney U test z = −0.487; *p* > 0.05.

In the results obtained by sex, we observe that in both cases the highest value in the answers was placed on the factor “Perception of the PE teacher’s behaviour”, among girls (M = 4.08 ± 0.72) and boys (M = 4.04 ± 0.69). When the two factors were compared by sex, not statistically significant differences were found (*p* > 0.05).

Table 4 presents the results for the mean scores obtained by the sample set as a function of the pupils motives for or against the practice of physical exercise (AMPEF), by grade. Considering the overall results of each grade, no statistically significant differences were found, with the following specific results in the Kruskal–Wallis test Chi^2^ = 1.608; gl = 2; *p* = 0.44.

It was also observed that the highest-valued factor was “Prevention and positive health”, among pupils in 8th grade primary; however, this level of interest was lower in higher years. When the factors were compared as a function of grade, no statistically significant differences were found for any factor, with specific results (*p* > 0.05).

Table 5 presents the results for the mean scores obtained in the sample set as a function of the pupils perception of the PE teacher’s behaviour, by grade. Considering the overall results of each grade, no statistically significant differences were found, with the following specific results of the Kruskal–Wallis test Chi2 = 3.306; gl = 2; *p* = 0.19.

It was also observed that the highest value placed on “Perception of the PE teacher’s behaviour” and “Perception of the contents of the PE class” was found among pupils in 1st grade secondary education. If we consider the differences found in the factors by grade, statistically significant differences were observed only in the “Perception of the PE teacher’s behaviour”, between pupils in 8th grade primary and those in 1st grade secondary, with specific results of (*p* < 0.05).

Figure 1 presents the results in the correlation between the variables “motivation for physical exercise” (ampef), and “perception of the climate in pe classes” (cappef). The analyses show a statistically significant association between the variables, and a weak positive correlation, with specific results in spearman’s coefficient of r = 0.317; *p* = 0.01. (source: own preparation).

## 4. Discussion

The study enabled us to identify pupils main motives for or against the practice of physical exercise and how they are related to perception of the behaviour of PE teachers and the contents of the PE programme. According to the results of the study, the principal motives for practising physical exercise are associated with prevention and positive health, fun and wellness, and improving physical qualities (muscular strength and endurance); this agrees with the results of other studies [36,37,38,39,40,41]. At the other end of the scale, the motives which generated the least interest in pupils in practising physical exercise were social recognition, and health emergencies. The same results have been obtained in other studies [39,40,42].

These results demonstrate pupils interests not only in practising physical exercise but also in PE class itself, leading to the conclusion that high percentages of pupils will continue to abandon sport [43,44] and PE classes. In general, it tends to be mostly girls who abandon PE classes, while Carrillo et al. [45] indicate that boys carry out more physical exercise in school time. Nevertheless, although girls are less committed to PE classes, they have a better understanding of the conceptual area of the discipline [46], and also obtain better academic results [47].

Turning to pupils perceptions of PE teachers behaviour and class contents, the results indicate that they are assessed positively, from which we may infer that the climate in PE classes is good; boys have a more positive perception of their teachers and class contents than girls. In the study by Moreno and Rodríguez [15], boys placed a lower value on the subject and the teacher than did girls. Another result observed in the present study is related to pupils perception of their teachers as a function of their grade; very positive responses were found in all grades, and no significant differences existed among grades. These results agree with those of Luke and Cope [48], who also found that attitudes towards teacher behaviour were positive in all grades. However, other studies [49,50] carried out in Chile show that pupils in higher grades have a more critical or unfavourable perception of the attitudes of their teachers. Therefore, it is very important to maintain an agreeable climate in the classroom, since this encourages satisfaction with the subject among pupils [51].

## 5. Conclusions

This study indicates that Chilean pupils have a high motivation to practice physical exercise, especially boys, who placed special emphasis on prevention and muscular strength and endurance; in girls the prevalent motives were prevention and body mass and body image.

Pupils perception of the attitudes of their PE teachers and the class contents are evidence of an excellent class climate and satisfaction with the class contents. Significant, weak correlation was found between pupils motivation to practice physical exercise and their perception of the climate in PE classes, so it is recommended that PE classes should be oriented by the interest shown by the pupils in different school contexts, stressing prevention and positive health.

### Practical Applications

Finally, the present work has important practical implications for the didactics of physical education. On the one hand it allows teachers to focus their planning as a function of the interests declared by pupils in practising physical exercise, while on the other it shows pupils validation of the behaviour of PE teachers and of the contents of the discipline. Teachers can therefore review their practices and intervention to ensure that they really contribute to a deeper understanding of an active, healthy life. Taking pupils interests into account can not only make the class a personal experience of an active life, but also contribute to bringing the social and cultural values of a society closer to living well and caring for the environment.

## Figures and Tables

**Figure 1 ijerph-18-08348-f001:**
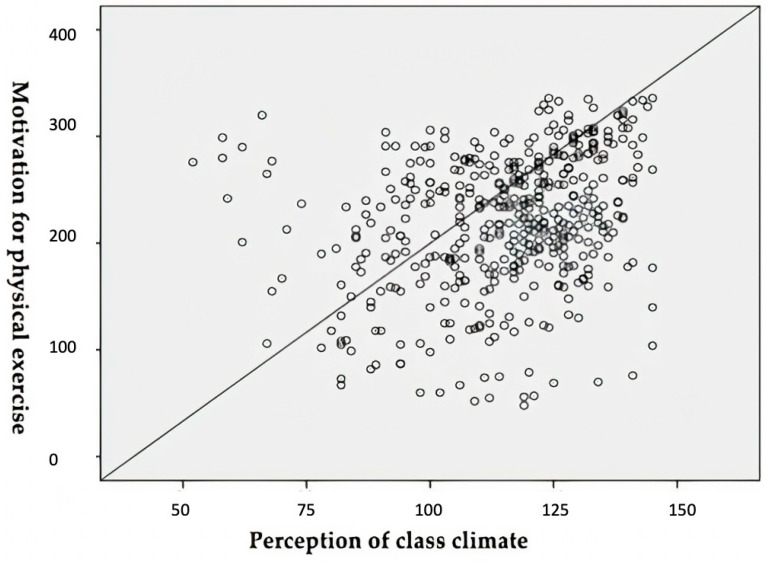
Correlation between AMPEF and CAPPEF.

**Table 1 ijerph-18-08348-t001:** Composition of the sample by grade and ages.

Grade	Frequency (ua)	Percentage (%)			
		Ages			
		16	17	18	19
8th grade primary	143	63	68	9	3
1st grade secondary	154	1	62	80	11
2nd grade secondary	151	4	72	75	151
Total	448	68	219	182	165

Source: Own preparation.

**Table 2 ijerph-18-08348-t002:** Descriptive results of the AMPEF factors, by sex.

	Girls	Boys	Total
Factors	M (SD)	M (SD)	M (SD)
Weight and body image	4.65 (1.49)	4.85 (1.60)	4.78 (1.57)
Fun and wellness	4.44 (1.73)	5.08 (1.52) *	4.84 (1.63)
Prevention and positive health	5.09 (1.40)	5.41 (1.46) *	5.29 (1.45)
Competition	3.53 (1.86)	4.22 (1.75) *	3.96 (1.82)
Affiliation	3.80 (1.73)	4.48 (1.62) *	4.23 (1.69)
Muscular strength and endurance	4.58 (1.60)	5.29 (1.49) *	5.03 (1.57)
Social recognition	3.42 (1.69)	3.95 (1.74) *	3.76 (1.74)
Stress control	4.50 (1.81)	4.55 (1.82)	4.53 (1.82)
Agility and flexibility	4.61 (1.73)	4.80 (1.69)	4.73 (1.71)
Challenge	4.46 (1.70)	4.95 (1.63) *	4.77 (1.67)
Health emergency	3.42 (1.70)	3.40 (1.63)	3.41 (1.66)
Total	4.23 (1.68)	4.63 (1.63) *	4.48 (1.68)

Note: * *p* < 0.05 statistically significant differences between girls and boys. Source: own preparation.

**Table 3 ijerph-18-08348-t003:** Descriptive results of the CAPPEF factors, by sex.

	Girls	Boys	Total
Factors	M (SD)	M (SD)	M (SD)
Perception of the PE teacher’s behaviour	4.08 (0.72)	4.04 (0.69)	4.06 (0.70)
Perception of the contents of the PE class	3.77 (0.62)	3.88 (0.57)	3.84 (0.59)
Total	3.93 (0.67)	3.96 (0.63)	3.95 (0.65)

Source: own preparation.

**Table 4 ijerph-18-08348-t004:** Descriptive results of the AMPEF factors, by grade.

	8th Grade Primary	1st Grade Secondary	2nd Grade Secondary	Total
Factors	M (SD)	M (SD)	M (SD)	M (SD)
Weight and body image	4.99 (1.43)	4.68 (1.52)	4.66 (1.72)	4.77 (1.56)
Fun and wellness	4.93 (1.58)	4.84 (1.55)	4.77 (1.74)	4.84 (1.62)
Prevention and positive health	5.44 (1.28)	5.22 (1.41)	5.21 (1.62)	5.29 (1.44)
Competition	4.20 (1.79)	3.82 (1.73)	3.89 (1.92)	3.97 (1.81)
Affiliation	4.38 (1.63)	4.20 (1.60)	4.12 (1.83)	4.23 (1.69)
Factor muscular strength and endurance	5.12 (1.44)	4.97 (1.52)	5.00 (1.73)	5.03 (1.56)
Social recognition	3.94 (1.79)	3.56 (1.62)	3.78 (1.80)	3.76 (1.74)
Stress control	4.39 (1.80)	4.57 (1.79)	4.63 (1.87)	4.53 (1.82)
Agility and flexibility	4.80 (1.68)	4.65 (1.64)	4.74 (1.80)	4.73 (1.71)
Challenge	4.86 (1.62)	4.75 (1.58)	4.70 (1.81)	4.77 (1.67)
Health emergency	3.47 (1.71)	3.41 (1.58)	3.35 (1.69)	3.41 (1.66)
Total	4.59 (1.61)	4.42 (1.59)	4.44 (1.78)	4.48 (1.66)

Source: own preparation.

**Table 5 ijerph-18-08348-t005:** Descriptive results of the CAPPEF factors, by grade.

	8th Grade Primary	1st Grade Secondary	2nd Grade Secondary	Total
Factors	M (SD)	M (SD)	M (SD)	M (SD)
Perception of the PE teacher’s behaviour	3.98 (0.73)	4.17 (0.67) *	4.02 (0.71)	4.06 (0.70)
Perception of the contents of the PE class	3.60 (0.53)	3.62 (0.58)	3.53 (0.62)	3.58 (0.58)
Total	3.79 (0.53)	3.90 (0.63)	3.78 (0.67)	3.82 (0.64)

Note: * *p* < 0.05. Source: own preparation.

## Data Availability

The underlying research materials related to this paper are available from the corresponding author upon request.

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
