# Peer review of "Motives of Students for or against the Practice of Physical Exercise and How They Are Related with the Climate in Physical Education Classes"

_ijerph, 2021, doi:10.3390/ijerph18168348_

Round 1

Reviewer 1 Report

Overall, I find the manuscript content/topic interesting. However, extensive English language editing is required for clarity and comprehension. There is much of the manuscript that is difficult to understand as written.

I suggest the authors double-check reference formatting. There are semicolons separating authors of the same publication (I believe these should be commas), making it difficult to differentiate between publications listed throughout the manuscript. Further, many publications are not in English, making it impossible to examine these publications for further research. 

I suggest the authors explain the levels (grades) of the students in the study; terminology is different in different countries so at least comparing "second grade secondary" to US or UK terminology is suggested. 

What are "direct grant schools"? 

I suggest that lines 136-142 be placed above the preceding paragraph, for clarity of IRB approval. 

Line 158-159 state Table 2 contains both AMPEF and CAPPEF results; it appears to only contain AMPEF results. 

Suggest consistent formatting for clarity and ease of reading. For example, the text in the "Boys" column in Table 2 is not all centered. 

I'm assuming "sample set" refers to the sample studied? 

Do the asterisks in Table 2 indicate gender differences? This is not clear by looking at the table. It is still unclear after reading the accompanying text.

Line 185 has p>0.05, is this correct? 

I suggest English labels for Figure 1

Use caution not to overstate your results. Your correlation was weak (as noted in the paper) and yet in the Discussion, the authors recommend PE classes be oriented by student interest. 

Author Response

Their suggestions were accepted and included in the article.

Reviewer 2 Report

The manuscript investigates the motives of students to complete Physical Education classes and their perceptions of the climate within the class with a specific focus on their perception of their teachers. The key findings are that the students are motivated to complete PE, with the most important factor being Prevention and positive health. Generally the manuscript is well-written and easy to follow. The conclusions match the findings, however there are a few minor changes required. Please see specific comments below:

Line 23: Replace 'relation' with 'correlation' 

Line 34: Rewrite to 'on the one hand it increases pupils' effort to learn...'

Line 95: Add more detail about how you removed outliers and cleaned the data.

Table 1: Add age data.

Line 122. Why did the experts decide to decrease the likert scale when the questionnaire was already validated. Justification needs to be provided.

Table 2: Columns need aligning.

Line 185: In the sentence you state statistically significant differences were found, whereas the P-value show they weren't. Which is it?

Line 195-196: 'from which we infer..' should be removed from the results as this is explanation and should be in the discussion.

Line 209. Give the actual P-value.

Figure 1: The axes titles need to be in English. The y-axis needs to start at 0. Is this the line of best fit or a line of identity?

Line 215. P<0.001 not P=0.00. P will never equal 0!

Line 230. Replace 'tension' with 'demonstrate'.

Line 260. Replace 'relations were' with 'correlation was'

Author Response

(The authors gave the same response as above.)

Reviewer 3 Report

Dear Authors

Consider all changes in order to improve the final document

Author Response

(The authors gave the same response as above.)

Round 2

Reviewer 3 Report

Dear Author:

In order to finish the process, last minor details

In Advanced

King Regards

1) LINE 228. TH in SUPERINDEX

2) LINE 240. Presents

3) References: Include DOI

49 LINE 450, a en superindex

Author Response

Dear Editor

changes to the manuscript were marked using the
"Track changes" function.

Kind regards
